# Physico-chemical characterization and topological analysis of pathogenesis-related proteins from *Arabidopsis thaliana* and *Oryza sativa* using *in-silico* approaches

Amritpreet Kaur[1], Pratap Kumar Pati[2]*, Aparna Maitra Pati[3]*, Avinash Kaur Nagpal[1]*

**1** Department of Botanical and Environmental Sciences, Guru Nanak Dev University, Amritsar, Punjab, India, **2** Department of Biotechnology, Guru Nanak Dev University, Amritsar, Punjab, India, **3** CSIR-Institute of Himalayan Bioresource Technology, Palampur, Himachal Pradesh, India

* avnagpal@yahoo.co.in (AKN); pkpati@yahoo.com (PKP); aparna@ihbt.res.in (AMP)

## Abstract

Plants are constantly under the threat of various biotic and abiotic stress conditions and to overcome these stresses, they have evolved multiple mechanisms including systematic accumulation of different phytohormones, phytoalexins and pathogenesis related (PR) proteins. PR proteins are cluster of proteins with low molecular weight which get incited in plants under different stresses. In this paper, *in-silico* approaches are used to compare the physico-chemical properties of 6 PR proteins (PR1, PR2, PR5, PR9, PR10, PR12) of *Arabidopsis thaliana* and *Oryza sativa*. Topological analysis revealed the presence of transmembrane localization of PR2 and absence of transmembrane domain in PR10 of both model plants studied. Amino acid composition shows the dominance of small aliphatic amino acids i.e. alanine, glycine and serine in both plants studied. These results highlights the similarities and differences between PRs of both model plants, which provides clue towards their diversified roles in plants.

## Introduction

Ever increasing human population and drastic climate change being observed in recent decades continue to pose serious threat to growth and productivity of agricultural crops. The latter encounter different types of environmental stresses, mainly categorized as biotic or abiotic. The abiotic stresses include salinity, cold, heat, drought, floods, heavy metals etc. whereas, biotic stresses include attack by pathogens (bacterial, fungal, viral). Both biotic and abiotic stresses are detrimental to plant growth and development because they are known to cause several metabolic dysfunctions in plants and in extreme cases can also cause death of the plant [1, 2]. During the course of evolution, plants have evolved a broad range of defense mechanisms for survival under various stressful conditions. These mechanisms involve the responses like activation of Reactive Oxygen Species (ROS); accumulation of different phytohormones like abscisic acid (ABA), ethylene (ET), jasmonic acid (JA), methyl jasmonate (MeJA) and salicylic

**Data Availability Statement:** All relevant data are within the manuscript. The study involves in-silico analysis of 6 PRs of Arabidopsis thaliana and Oryza

sativa. In the analysis, the data was tabulated from annotations of Uniprot files of each PR and their accession numbers are given in Table 1.

**Funding:** The author(s) received no specific funding for this work.

**Competing interests:** The authors have declared that no competing interests exist.

acid (SA); production of pathogenesis related (PR) proteins and accumulation of phytoalexins. Out of these, PR protein production and accumulation in plants during biotic and abiotic stresses is very crucial. They are not only induced by different stresses but also accumulate in plant tissues during different developmental stages like flowering, senescence etc.. First PR protein was isolated from tobacco leaves but now its presence is reported in many plants. Initially PR proteins were classified into 17 families based on their serological relationships, biological or enzymatic activity and sharing of amino acid sequences [3]. Among them, many PRs are antifungal (PR1, PR2, PR3, PR4, PR5, PR7, PR12, PR13 and PR14); a few possess endochitinase activity (PR8 and PR11); whereas two others are oxalate oxidases (PR15 and PR16). PR17 proteins are basic secretary proteins [4–7]. Custers et al., [8] characterized a class of carbohydrate oxidases with a defensive role in higher plants, later on these were categorized as PR18 [9]. Sooriyaarachchi et al., [10] purified new antimicrobial protein from *Pinus Sylvestris* and categorized them as PR19. The present study is focused on PR1, PR2, PR5, PR9, PR10 and PR12; with diverse functions as mentioned below.

PR1 is the first discovered class of pathogenesis related proteins with molecular weight ranging from 14–17 KDa. The first PR1 protein was isolated from tobacco after which PR1 proteins were identified in number of plant species including Arabidopsis, barley, maize, pepper, tomato, wheat, rice etc. [11]. Gamir et al., [12], for the first time described mode of action of PR1 proteins which were shown to inhibit the sterol-auxotrophic pathogenic microorganisms by binding and sequestering sterols. PR1 proteins are known to have antifungal activity and their increased expression is considered as hallmark for activation of SA mediated signaling pathway [11, 13]. PR2 proteins are low molecular weight proteins in the range of approximately 6–43 KDa. They have been identified in various plants species such as tomato, barley, rice etc. [14–16]. PR2 proteins are strong anti-fungal proteins, possessing β-1, 3-glucanase activity, catalyzing hydrolytic cleavage of β-1,3-glucosidic linkages present in the β-1,3-glucan, resulting in breakdown of fungal cell wall [17]. They have role not only in providing defense against the pathogen attack but also in different developmental processes like pollen germination, embryogenesis, fruit ripening, seed germination and development etc. [18]. PR5 proteins are thaumatin like proteins (TLPs) because of their sequence similarity with thaumatin, which is sweet protein from *Thaumatococcus danielli*, a west African shrub. They have diverse roles in managing biotic and abiotic stresses and have antifungal, antipest, antifreeze activities and also exhibit tolerance to osmotic stress [19]. PR9 proteins are peroxidases with molecular weight ranges from 32–42 KDa involved in lignification of plant cell wall [20]. PR10 proteins are ribonuclease like proteins having antibacterial activity against number of bacterial species including those of *Pseudomonas*, *Agrobacterium*, *Serratia* etc. [3, 21, 22]. They are also known to possess antiviral activity. Their role in abiotic stress management has been demonstrated in rice and maize [23, 24]. PR12 (defensins) are small antimicrobial peptides with very low molecular weight ranging from 3–5 KDa. They provide protection against wide range of microbial pathogens including bacteria and fungi. They have role in management of abiotic stress like cold, drought, heavy metals etc. [25]. Many PR proteins are induced through the action of many phytohormones like JA, SA and ET and the level of these hormones is also known to increase at the site of infection in a plant [6]. It is clear that PR proteins are associated with resistance against various kinds of stresses (abiotic and biotic); and plant development and morphogenesis related signaling pathways but the mechanism of their action is yet to be elucidated. Hence, the detailed analysis of their varied physico-chemical properties could throw light on their useful multiplicity.

Apart from different experimental methods, many *in-silico* methods and online tools are available for the analysis and characterization of protein sequences. Bioinformatics tools offer researchers a quick and cost effective information on the physico-chemical properties of

proteins that is useful in planning laboratory experiments.. Different physico-chemical properties like protein length, amino acid composition, molecular weight, aliphatic index, extinction coefficient, isoelectric point, half-life, instability index and grand average of hydropathicity can be analyzed. Several studies have shown the presence and expression of PR genes in various plants including *Arabidopsis thaliana* and *Oryza sativa*. In our earlier study, we compared cis-elements in the promoter regions of PR proteins using *in-silico* tools [26]. To the best of our knowledge, there is no report on physicochemical properties and topology of various PRs of *A. thaliana* and *O. sativa*. Hence, the present study was planned to analyze and compare physicochemical properties and topology of 6 PR (PR1, PR2, PR5, PR9, PR10 and PR12) proteins of each of the two model plants. This study will aid in understanding the occurrence of diversification in different PR proteins of *A. thaliana* and *O. sativa*. Further, it will also throw light on the similarities and differences between *A. thaliana* and *O. sativa* PR protein sequences.

## Materials and methods

### Retrieval of protein sequences

Six PR protein sequences viz. PR1, PR2, PR5, PR9, PR10 and PR12 of *A. thaliana* and *O. sativa* were downloaded from UniProt (http://www.uniprot.org) [27] in FASTA format for further analysis.

### Analysis of physico-chemical properties of PRs

Physico-chemical properties of PR1, PR2, PR5, PR9, PR10 and PR12 of *A. thaliana* and *O. sativa* were computed using ExPASy ProtParam tool (https://web.expasy.org/protparam/) [28]. Cellular localization of PR proteins was predicted by using web servers like: CELLO v.2.5 (http://cello.life.nctu.edu.tw/) [29]; WoLF PSORT (http://wolfpsort.seq.cbrc.jp/) [30]; and EuLoc (http://euloc.mbc.nctu.edu.tw/) [31]. For detection of signal peptides, PrediSi (http://www.predisi.de/) [32] and SignalP 4.1 Server (http://www.cbs.dtu.dk/services/SignalP/) [33] were used.

### Topological analysis

Topological analysis of each PR protein of *A. thaliana* and *O. sativa* was carried out using online tools viz. TMpred (http://www.ch.embnet.org/software/TMPRED_form.html) [34]; TMAP (http://www.bioinformatics.nl/cgi-bin/emboss/tmap) [35]; PHDhtm (https://npsa-prabi.ibcp.fr/cgi-bin/npsa_automat.pl?page=/NPSA/npsa_htm.html) [36]; DAS (http://www.sbc.su.se/~miklos/DAS/) [37], HMMTOP (http://www.enzim.hu/hmmtop/) [38]; TMHMM (http://www.cbs.dtu.dk/services/TMHMM/) [39]; Phobius (http://phobius.sbc.su.se/) [40]; WHAT (http://saier-144-21.ucsd.edu/barwhat.html) [41] and MEMBRAIN (http://www.csbio.sjtu.edu.cn/bioinf/MemBrain) [42].

### Secondary structure prediction

Secondary structures of PR proteins were predicted with online Expasy SOPMA tool (Self-Optimized Prediction Method and Alignment) (https://npsa-prabi.ibcp.fr/cgibin/npsa_automat.pl?page=/NPSA/npsa_sopma.html) [43]. This tool provides details about different conformations of proteins from the given sequences such as percentages of α-helices, β-sheets, turns, extended strands and random coils.

**Table 1. Physico-chemical properties of Pathogenesis Related proteins of *A. thaliana* and *O. sativa* computed using ProtParam tool.**

| PR | Accession no. | | Length | | M.Wt. | | pI | | (-) R | | (+) R | | Ec$^a$ | | II | | AI | | GRAVY | |
|----|------|------|-----|-----|----------|----------|------|------|-----|-----|-----|-----|-------|-------|-------|-------|-------|-------|--------|--------|
| | At | Os | At | Os | At | Os | At | Os | At | Os | At | Os | At | Os | At | Os | At | Os | At | Os |
| PR1 | P33154 | Q6YSF8 | 161 | 168 | 17676.94 | 17534.13 | 9 | 4.55 | 10 | 17 | 16 | 8 | 38765 | 43930 | 25.57 | 46.09 | 73.85 | 59.94 | -0.288 | -0.176 |
| PR2 | P33157 | Q94CR1 | 339 | 334 | 37338.92 | 35682.43 | 4.85 | 9.40 | 39 | 23 | 27 | 31 | 41830 | 40800 | 48.07 | 31.95 | 81.68 | 85.96 | -0.266 | -0.061 |
| PR5 | P50700 | Q7XST4 | 244 | 278 | 26632.76 | 29863.03 | 6.08 | 7.98 | 18 | 21 | 17 | 24 | 34920 | 42285 | 44.2 | 46.81 | 59.63 | 63.53 | -0.295 | 0.040 |
| PR9 | P0DI10 | Q7F1U0 | 325 | 317 | 35624.39 | 42889.77 | 9.35 | 5.77 | 31 | 23 | 44 | 20 | 23420 | 11960 | 35.32 | 30.35 | 97.17 | 83.56 | -0.049 | 0.028 |
| PR10 | Q93VR4 | Q9LKJ9 | 155 | 158 | 17045.5 | 16656.91 | 5.09 | 4.95 | 23 | 24 | 17 | 16 | 19940 | 16180 | 21.89 | 30.69 | 92.45 | 83.42 | -0.163 | -0.118 |
| PR12 | P30224 | Q6K209 | 80 | 80 | 8709.22 | 8784.33 | 8.47 | 9.08 | 4 | 6 | 7 | 12 | 8980 | 500 | 27.49 | 47.84 | 75.5 | 65.88 | 0.339 | -0.041 |

At = *A. thaliana* Os = *O. sativa*.

**M.wt.** = Molecular weight; **pI** = Isoelectric point; **(-) R** = total number of negatively charged residues; **(+) R** = total number of positively charged residues; **Ec$^a$** = Extinction coefficient ($^a$units of M$^{-1}$ cm$^{-1}$ at 280 nm measuring in water); **II** = Instability index; **AI** = Aliphatic index; **GRAVY** = Grand average of hydropathicity.

## Results

The results on different PR proteins of *Arabidopsis thaliana* and *Oryza sativa* viz. PR1, PR2, PR5, PR9, PR10 and PR12 with respect to their accession numbers, physico-chemical properties, cellular localization, topology and signal peptides are presented in Tables 1–3.

### Physico-chemical properties

Physico-chemical properties like protein length, molecular weight, isoelectric point (pI), total number of negatively and positively charged residues, extinction coefficient, instability index (II), aliphatic index (AI) and grand average of hydropathicity (GRAVY) for PRs of *A. thaliana* and *O. sativa* were computed using ExPASy ProtParam tool (Table 1). Among all PRs of both species; *A. thaliana* and *O. sativa*, minimum length was observed in AtPR12 and OsPR12 (80) while maximum length was observed for PR2 (339 in AtPR2 and 334 in OsPR2). Accordingly minimum molecular weight for PRs of both plants was observed for PR12 (8709.22 for AtPR12 and 8784.33 for OsPR12) whereas, maximum molecular weight was observed for PR2 (37338.92) in case of *A. thaliana* and PR9 (42889.77) for *O. sativa*. On the basis of computed

**Table 2. Subcellular localization of PR proteins predicted by different tools.**

| Tool PR | | Cello | Euloc | Wolfpsort |
|---------|-----|--------------|--------------|----------------------|
| PR1 | At | Extracellular | Extracellular | Extracellular/Vacuole |
| | Os | Extracellular | Vacuole | Extracellular |
| PR2 | At | Vacuole | Vacuole | Vacuole/Extracellular |
| | Os | Vacuole | Vacuole | Chloroplast |
| PR5 | At | Extracellular | Vacuole | Chloroplast |
| | Os | Extracellular | Extracellular | Extracellular |
| PR9 | At | Extracellular | Extracellular | Extracellular |
| | Os | Extracellular | Extracellular | Chloroplast |
| PR10 | At | Cytoplasmic | Cytoplasmic | Cytoplasmic |
| | Os | Cytoplasmic | Cytoplasmic | Cytoplasmic |
| PR12 | At | Extracellular | Extracellular | Extracellular |
| | Os | Extracellular | Extracellular | Extracellular |

At: *Arabidopsis thaliana*; Os: *Oryza sativa*.

**Table 3. Comparative topological analysis of PR proteins of *A. thaliana* and *O. sativa* indicating the presence or absence of transmembrane domains using different online tools.**

| PR protein TOOL | PR1 | | PR2 | | PR5 | | PR9 | | PR10 | | PR12 | |
|---|---|---|---|---|---|---|---|---|---|---|---|---|
| | At | Os | At | Os | At | Os | At | Os | At | Os | At | Os |
| TMHMM | 0 | 0 | 1 | 1 | 1 | 0 | 1 | 1 | 0 | 0 | 1 | 0 |
| Phobius | 0 | 0 | 1 | 1 | 0 | 1 | 1 | 1 | 0 | 0 | 0 | 0 |
| HMMTOP | 1 | 0 | 1 | 1 | 1 | 0 | 1 | 1 | 0 | 0 | 1 | 1 |
| TMpred | 1 | 1 | 1 | 1 | 1 | 1 | 1 | 1 | 0 | 0 | 1 | 1 |
| PHDhtm | 1 | 1 | 1 | 1 | 1 | 1 | 1 | 1 | 0 | 0 | 1 | 1 |
| TMAP | 1 | 1 | 1 | 1 | 1 | 1 | 1 | 1 | 1 | 0 | 1 | 1 |
| DAS | 1 | 1 | 1 | 1 | 1 | 1 | 1 | 0 | 0 | 0 | 1 | 1 |
| MEMBRAIN | 1 | 0 | 1 | 1 | 0 | 1 | 0 | 1 | 0 | 0 | 1 | 1 |
| Various tools predicting the percentage of transmembrane domains | 75 | 50 | 100 | 100 | 75 | 75 | 87.5 | 87.5 | 0 | 0 | 87.5 | 75 |

At: *Arabidopsis thaliana*; Os: *Oryza sativa* (0-absent; 1-Present).

pI values, three PR proteins of *A. thaliana* (AtPR2, AtPR5 and AtPR10) were found to be acidic while other three (AtPR1, AtPR9 and AtPR12) were basic in nature whereas in case of *O. sativa*, OsPR1, OsPR9 and OsPR10 were acidic and OsPR2, OsPR5 and OsPR12 were basic in nature. In case of AtPRs, number of positively charged residues were more as compared to negatively charged residues for PR1, PR9 and PR12, whereas, in case of OsPRs, more positively charged residues were observed for PR2, PR5 and PR12. Extinction coefficient (EC) values ranged from 8980 (AtPR12) to 41830 $M^{-1}$ $cm^{-1}$ (AtPR2) for AtPRs and 500 (OsPR12) to 43930 $M^{-1}$ $cm^{-1}$ (OsPR1) for OsPRs. Instability index of four PRs of *A. thaliana* (AtPR1, AtPR9, AtPR10 and AtPR12) and three PRs of *O. sativa* viz. OsPR2, OsPR5 and OsPR12 was less than 40, which means that they are stable. AI values ranged from 59.63 (AtPR5) to 97.17 (AtPR9) for *A. thaliana* PR proteins and 59.94 (OsPR1) to 85.96 (OsPR2) for *O. sativa* PR proteins. GRAVY score of five (PR1, PR2, PR5, PR9, PR10) of the six PR proteins analyzed in *A. thaliana* was found to be negative with range from -0.049 to –0.295 with the exception of AtPR12 whose GRAVY score was 0.339. GRAVY score of OsPR proteins was found to be negative (-0.061 to -0.176) for OsPR2, OsPR12, OsPR10, OsPR1 and score was positive for OsPR5 (0.040) and OsPR9 (0.028). Amino acid composition of PR proteins of *A. thaliana* and *O. sativa*, with respect to aromatic, polar, non-polar, positively and negatively charged amino acids, is given in Fig 1. S1 Fig shows the comparison of percentage of each amino acid in different PR proteins of *A. thaliana* and *O. sativa*. Heat map representation of the comparison of amino acid composition of PR proteins of *A. thaliana* and *O. sativa* is shown in Fig 2.

Three different tools (Cello, Euloc and Wolfpsort), used to determine subcellular localization of different PR proteins in *A. thaliana* and *O. sativa*, revealed the extracellular localization of PR12 in both *A. thaliana* and *O. sativa*, PR5 in *O. sativa* and PR9 in *A. thaliana*. PR10 protein was found to be localized in cytoplasm of both the plants as predicted by all the three tools. Variable results were observed for PR1 and PR2 in both *A. thaliana* and *O. sativa*; PR5 in *A. thaliana* and PR9 in *O. sativa*, with regard to their localization (Table 2). The tools, PrediSi and SignalP 4.1 predicted the presence of signal peptides in all PR proteins of *A. thaliana* and *O. sativa* except PR10 protein i.e. AtPR10 and OsPR10.

## Topological analysis

Comparative topological analysis of different PR proteins of *A. thaliana* and *O. sativa* using various online tools indicated the presence of transmembrane domains in all PR proteins

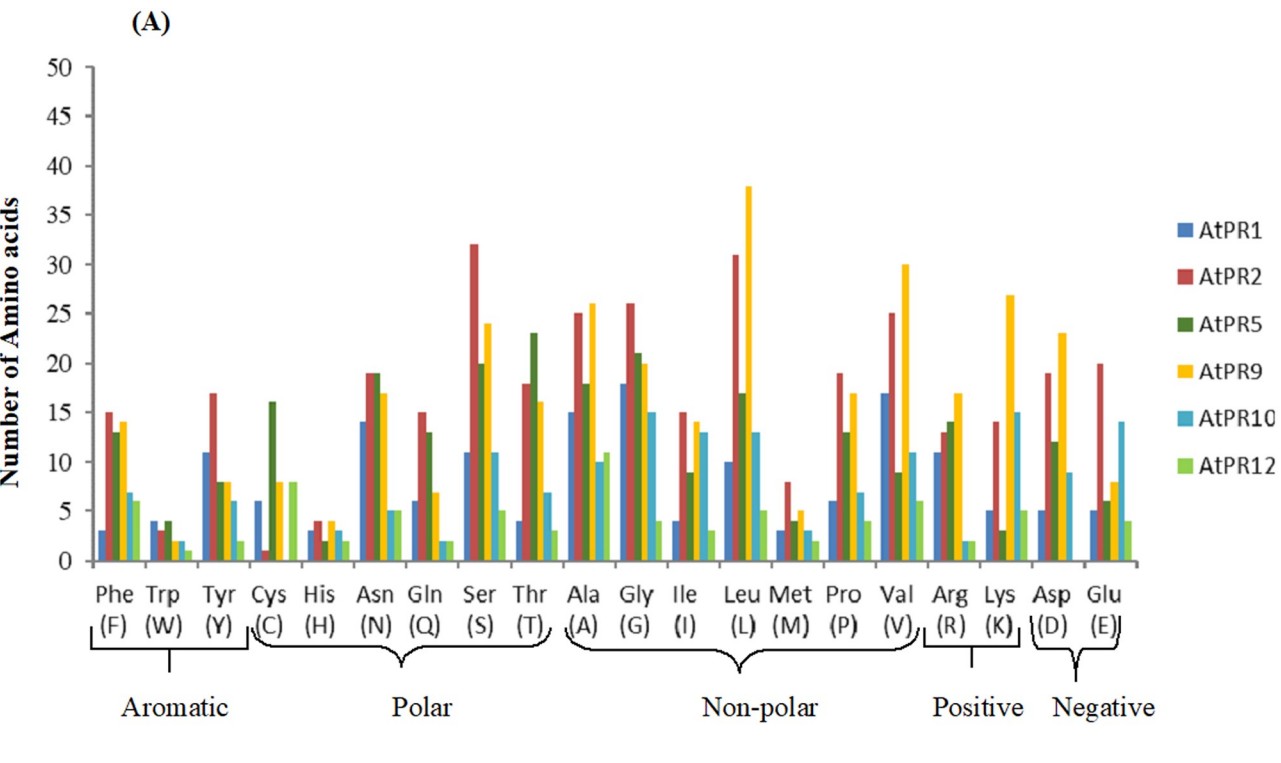

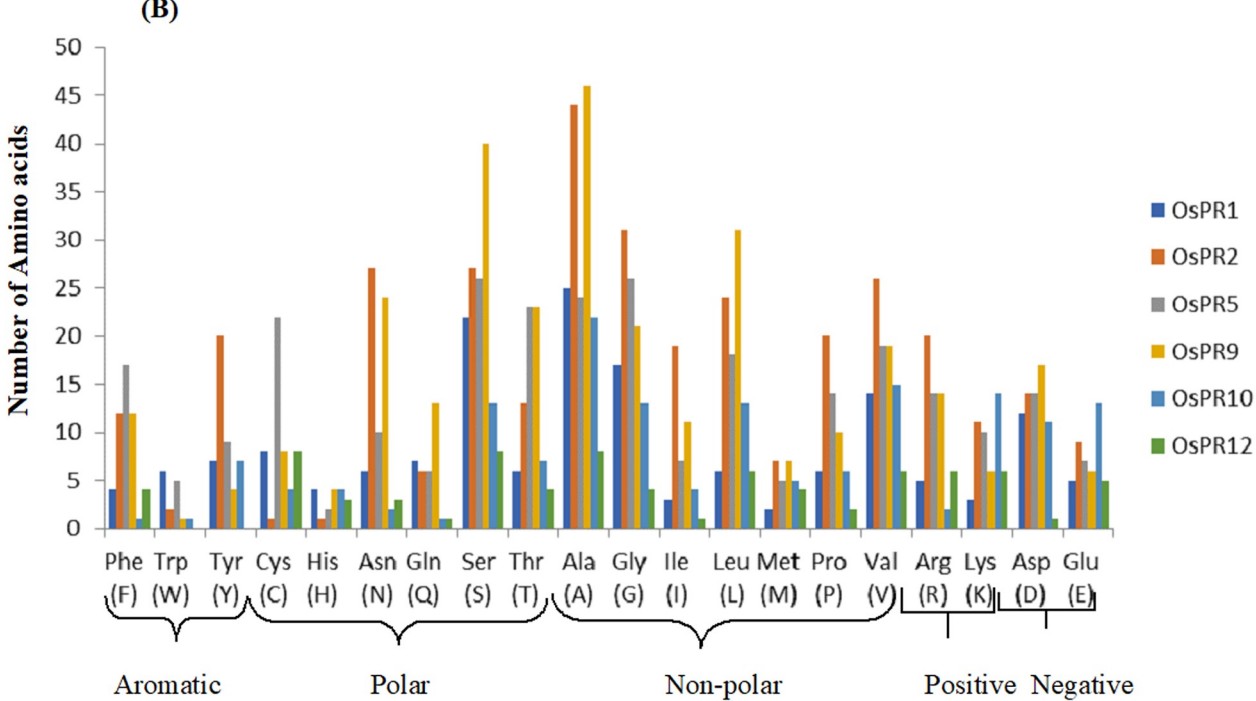

**Fig 1. Comparison of amino acid composition of PR proteins of *A. thaliana* (A) and *O. sativa* (B) with respect to their number of each amino acid in their respective class.**

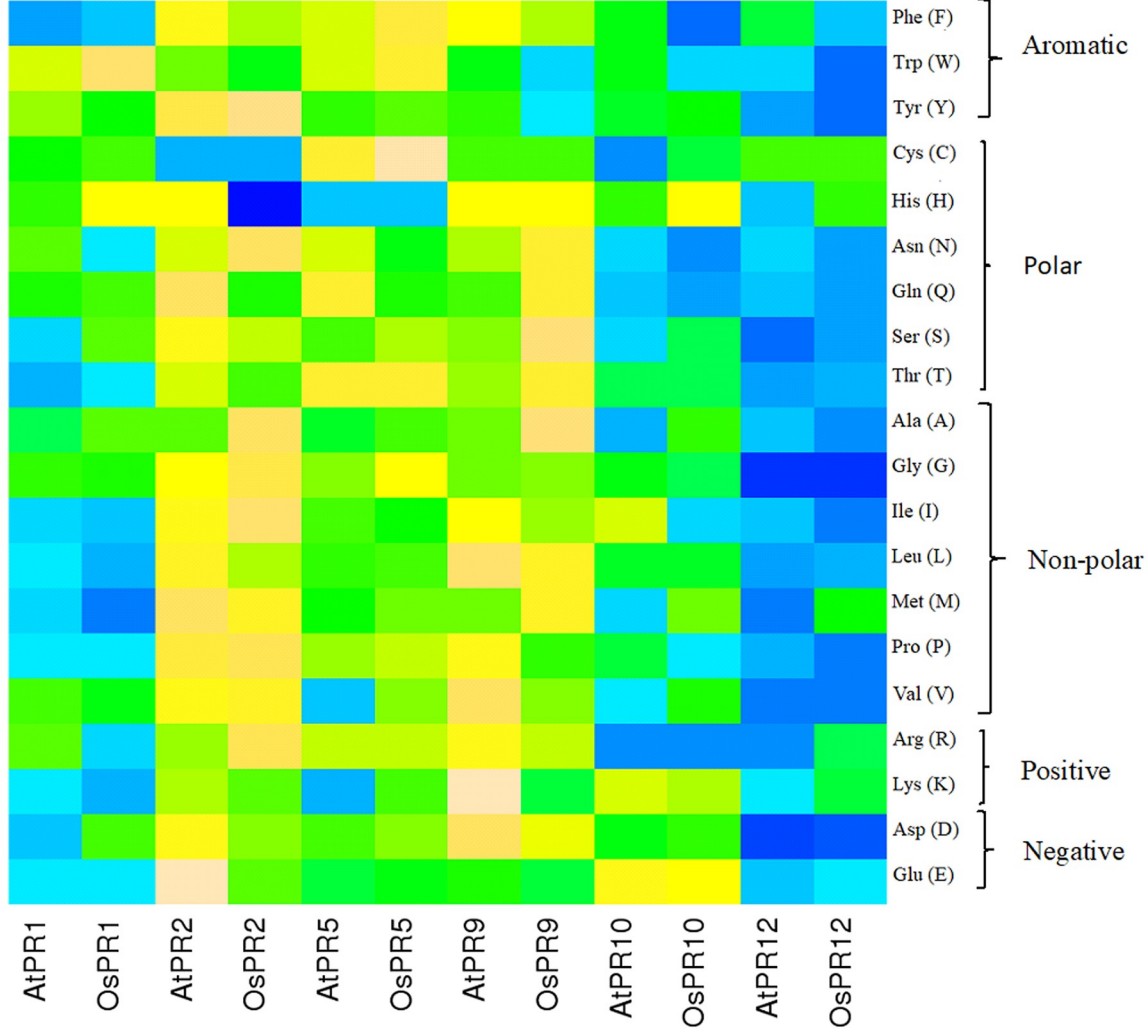

**Fig 2. Heat map representation showing the comparison of amino acid composition of PR proteins of *A. thaliana* and *O. sativa*.**

studied except PR10 (Table 3). 100% of the tools used in the study predicted the presence of transmembrane domains in PR2, whereas, 87.5% tools for PR9 and 75% for PR5 indicated the presence of transmembrane domains for both species, *A. thaliana* and *O. sativa*.

## Secondary structure prediction

SOPMA tool was used to predict percentage occurrence of secondary structure features (alpha helices, extended strands, beta turns and random coils) of PR proteins of *A. thaliana* and *O. sativa* (Figs 3 and 4).The analysis revealed occurrence of maximum frequency of random coils in PR1 and PR5 of both *A. thaliana* and *O. sativa*, whereas, alpha helices were found to be maximum in PR2, PR9 and PR12 proteins of both the plants. In case of PR10 protein, number

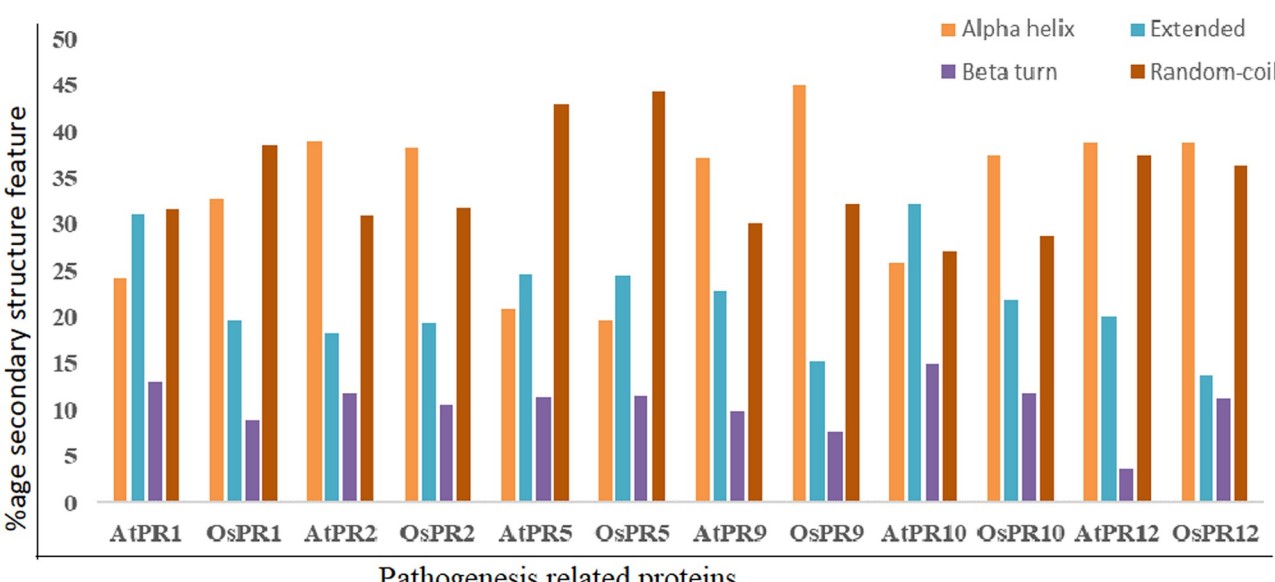

**Fig 3. Percentage occurrence of secondary structural features.**

of extended strands was maximum in *A. thaliana*, whereas alpha helices were found to be maximum in *O. sativa*.

## Discussion

The present study focused on *in-silico* analysis of physico-chemical properties of 6 PR proteins (PR1, PR2, PR5, PR9, PR10, PR12) w.r.t various parameters including their subcellular localization, topology and detection of signal peptides. Comparison of protein length and molecular weight of each type of PR in both species (*A. thaliana* and *O. sativa*) showed little variation. However, protein lengths and molecular weights of different PRs (PR1, PR2, PR5, PR9, PR10 and PR12) within each plant were significantly different. Subcellular localization, interactions and solubility depend upon isoelectric point and number of positively and negatively charged residues. pI is the pH value at which proteins carry no charge or the sum of negatively and positively charges is equal. pI value more than 7 was observed for PR12 of both the species; PR1 and PR9 of *A. thaliana*; and PR2 and PR5 of *O. sativa*. For PR10 of both the plants, the pI value was less than 7. This study is in line with some of the earlier studies which show that PR1 and PR2 can either be acidic or basic in nature [6, 44]. The acidic nature of PR10 observed for both the species studied is in confirmation with an earlier study which also showed PR10 to be acidic [45]. The EC value of a protein solution is an important parameter based on amount of light absorbed per mole of protein at a certain wavelength, most commonly 280 nm wavelength is used. EC value of protein is calculated from the number of tryptophan, tyrosine and cysteine residues per molecule because these residues contribute significantly to measured optical density of denatured protein at 276–282 nm range [28, 46, 47]. In the present study, minimum EC value (500 $M^{-1}$ $cm^{-1}$) was observed for OsPR12 which is mainly due to four cystine amino acids (two cysteines joined by disulphide bond form cystine). For, AtPR12 relatively higher value of EC (8980 $M^{-1}$ $cm^{-1}$) was observed which is due to the presence of 1 tryptophan and 2 tyrosine residues in addition to four cystines (8 cysteines). Among different PR proteins analysed, PR2 of both the species was found to be tyrosine rich and PR5 was cystine rich.

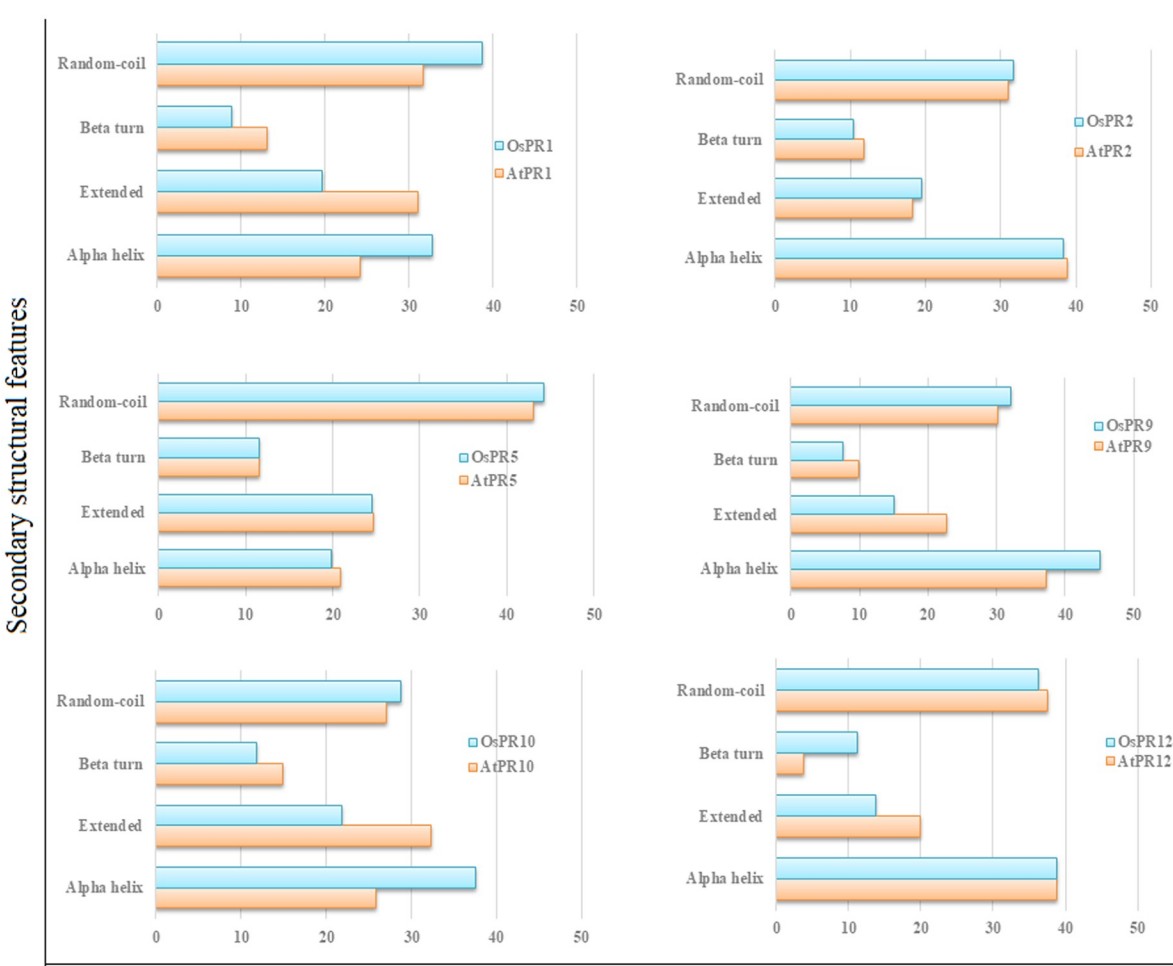

**Fig 4. Comparison of secondary structure features of PR proteins of *A. thaliana* and *O. sativa*.**

Instability index (II) indicates about the protein stability under both *in-vivo* and *in-vitro* conditions. Proteins with instability index (II) <40 are considered to be stable and those with II value >40 are referred to as unstable [48]. Instability index of PR9 and PR10 of both species studied; and PR1 and PR12 of *A. thaliana* only were found to be less than 40 indicating their stable nature. PR5 of both plants; PR2 of *A. thaliana* only; PR1 and PR12 of *O. sativa* only had stability index of more than 40, indicating them to be unstable proteins. Apart from instability index (II), aliphatic index (AI) is another parameter to check the stability of the proteins. For a protein, AI can be defined as the relative volume captured by aliphatic side chains of amino acids like A (alanine), V (valine), L (leucine) and I (isoleucine). Earlier a good correlation was established between AI and thermostability of proteins by Ikai [49]. Among the PRs of *A. thaliana* and *O. sativa*, AtPR5 (59.63) and OsPR1 (59.94) have lower values of AI as compared to other PRs of *A. thaliana* and *O. sativa*; indicating that they are less thermostable and have more flexible protein structure. The high AI value indicates that under wide range of temperature conditions, the protein is stable. Apart from studying protein concentration and stability, its hydrophobic or hydrophilic character is also analyzed with the GRAVY score. GRAVY score for particular protein is calculated as the sum of hydropathy values of all amino acids

present in the protein, divided by the number of residues in that protein. Its value lies between -2 to +2 where; negative score means hydrophilicity and positive score indicates hydrophobicity [50]. Proteins with more negative GRAVY score are considered to be hydrophilic in nature with good solubility and vice-versa. If a protein has GRAVY score more than 0.4, suggest its hydrophobic nature and difficult to detect on 2-D gels [51].

*In-silico* approaches have been used to determine subcellular localization of proteins which plays an important role to depict their function. Three different tools viz., Cello, Euloc and Wolfpsort were used to determine subcellular localization of different PR proteins analysed in this study. Cello tool is a two level support vector machine (SVM) classifier system and its prediction regarding subcellular localization for a particular protein is considered accurate/acceptable if its reliability/confidence value is at least 1 [52]. In our study, we observed minimum confidence score of 1.5 in case of AtPR9 and 2.9 for OsPR10. Whereas, maximum score of 4 was obtained for AtPR1, OsPR1, OsPR5, OsPR9 and OsPR12. Euloc is a hybrid tool that integrates three different approaches like homology search, Hidden Markov Model and SVM for detection of subcellular localization of proteins [31]. Wolfpsort predicts multi-site (nine sites) localization of a protein. It is a Sequence-based prediction method which along with homology/ functional motifs and sorting signals has greatly improved the accuracy of the prediction of subcellular localization of proteins [53].

Subcellular localization of PR proteins as predicted by Cello and Euloc was similar for all the PR proteins of *A. thaliana* and *O. sativa* except for PR1 of *O. sativa* and PR5 of *A. thaliana*. The results obtained by Wolfpsort for PR1, PR2, PR5 and PR9 were different from either predicted by Euloc and Cello or both. Subcellular localization of PR10 or PR12 was found to be cytoplasmic and extracellular, respectively and was uniformly predicted by all the three tools used. PR1 is an important antifungal protein and is known to be localized in extracellular space. Previously the localization of PR1 is detected in vacuoles, vesicles of cortical cytoplasm, Endoplasmic reticulum bodies etc., using prolonged dark incubation in combination with salicylic acid treatment of seedlings of *A. thaliana* by Pecenkov et al., [54]. In our study also, this protein (PR1) was shown to be localized in vacuoles (Euloc in *O. sativa* and Wolfpsort in *A. thaliana*). PR2 are group of proteins involved in number of developmental processes as well as in defense against biotic stress. Many studies reported the difference in localization of PR2 proteins in potato cultivars susceptible or resistant to PVY infections. The level of PR2 was found to be higher in cell walls, chloroplasts and vacuoles of susceptible cultivar [55]. In a number of in vivo studies, PR5 proteins have shown anti-microbial activities [56, 57]. PR5 has been shown to exhibit sequence similarity with a sweet tasting protein, thaumatin, which is derived from *Thaumatococcus daniellii*, a shrub from South Africa. Hence, they are also called as Thaumatin-like proteins (TLPs) [58]. Though PR5 proteins are called as TLPs, but none of them have a sweet taste like thaumatin. Based on their molecular weight, TLPs have been shown to fall into two categories viz. high molecular weight group with molecular weight range between 22–26 kDa and low molecular weight group with range less than 18 kDa. High molecular weight TLPs have been shown to get accumulated in cell vacuoles whereas, low molecular weight TLPs are extracellular [59]. PR5 of *Gossypium hirsutum* (GhPR5) has been shown to contain 242 amino acids with a signal peptide at the N-terminal end that facilitates its secretion into the extracellular space and a signal peptide at C-terminal signal to transport it to vacuoles [60]. In our study, PR5 proteins were found to be of high molecular weight with 26 kDa for *A. thaliana* and 29 kDa for *O. sativa*. Though being of high molecular weight PR5 of *O. sativa* was predicted to be extracellular by three different tools used in the study. However, one of the tools, Euloc indicated the localization of PR5 protein in the vacuoles of *A. thaliana*. The extracellular location of PR5 is indicative of presence of signal peptide in the N-terminal end of *O. sativa* as well as in *A. thaliana* as predicted by Cello tool. The vacuolar location of

PR5 in *A. thaliana* as predicted by Euloc tool is the indication of presence of signal peptide at C-terminal end of protein. PR9 are peroxidases and provide resistance against pathogens. They are extracellular or transmembrane proteins playing an important role in plant cell wall construction by catalyzing lignification [61]. Earlier the subcellular localization of peroxidases from sweet potato (*swpa4*) using fluorescence microscopy was evaluated, and observed the expression of *swpa4* in the extracellular space of cell [62]. PR10 are localized in the cytoplasm and are non-transmembrane proteins getting induced during several biotic and abiotic stresses [63]. In our study, we also found localization of PR10 in the cytoplasm of *A. thaliana* and *O. sativa*. Defensins are small and globular proteins belong to PR12 type of PR proteins present in extracellular spaces of plant cells. They are known to provide first line of immunity against pathogen attack [64, 65]. All the three tools used in the present study revealed the extracellular localization of PR12 of both *A. thaliana* and *O. sativa*.

For the topological analysis, 8 different tools were used (Table 3). Out of these 8 tools, 3 tools viz TMpred, PHDhtm and TMAP gave similar results with respect to presence or absence of transmembrane domains for all the 6 PR proteins in both the species. Whereas, other tools showed variable results for one or the other PR protein. The tool TMHMM indicated the presence of transmembrane domain in AtPR5 as well as AtPR12 but not in OsPR5 and OsPR12. This might be due to some prediction error of the tool itself [66] or may be the transmembrane domain of OsPR5 and OsPR12 did not meet the cutoff of the tool [67].

Analysis of amino acid composition of all PR proteins of *A. thaliana* and *O. sativa* revealed the dominance of amino acids with small aliphatic side chains (alanine, serine, glycine and threonine). Glycine is the smallest amino acid without any side chains and is often found in loop regions. The frequent presence of glycine has been reported in membrane proteins mainly in the transmembrane helices, suggesting its structural role [68, 69]. In the present study, among different PR proteins maximum percentage of glycine was observed for PR1 which has been shown to be transmembrane protein by 6/8 tools in *A. thaliana* and 4/8 tools in *O. sativa*. PR10 protein has been localized in cytoplasm by three software's used in the study. However, it also shows high percentage of glycine amino acid in both plants. Alignment of PR10 of *A. thaliana* and *O. sativa* revealed the presence of glycine motif (GXGGXG), which is also known as RNA binding site. The glycine motif has been shown to be involved in different enzymatic processes, membrane binding and transport, biosynthesis of secondary metabolites, binding to phytohormones etc [70]. Side chain of alanine being non-reactive is not directly involved in the function of the protein but has significant role in substrate recognition. Serine and threonine have a fairly reactive hydroxyl group which forms hydrogen bonds with number of polar substrates. Serine forms a catalytic triad along with histidine and aspartic acid (Asp-His-Ser) in many hydrolases. In rare cases, serine is replaced by cysteine in catalytic triad to fulfil same role. Proline has been shown to act like a molecular chaperon which provide protection against abiotic and biotic stresses by enhancing activities of some enzymes as well as maintaining integrity of proteins [71].

## Conclusion

PR proteins are defense related inducible proteins associated with resistance to various kinds of biotic and abiotic stresses in plants. In the present study, several bioinformatics tools were used to study the variations in the physicochemical properties and topology of 6 PRs each of *A. thaliana* and *O. sativa*. The results of this study demonstrated that PR2 protein of both *A. thaliana* and *O. sativa* are the larger proteins with molecular weights of 37KDa and 35KDa, respectively followed by PR9, in both species viz. *A. thaliana* (35KDa) and *O. sativa* (42 KDa). Among *A. thaliana* and *O. sativa* PRs, maximum AI was observed for AtPR9 (97.17) and

OsPR2 (85.96), respectively. For the subcellular localization prediction, we used 3 tools viz. Cello, Euloc and Wolfpsort. All these tools gave similar results for almost all of the PRs except for AtPR5, OsPR1 and OsPR2. This study throws light on the similarities and differences among the physio-chemical properties, topology, amino acid composition and secondary structure features of 6 PRs (PR1, PR2, PR5, PR9, PR10, PR12) of *A. thaliana* and *O. sativa*. This study will help in understanding the occurrence of diversification and functional multiplicity of various PR proteins of *A. thaliana* and *O. sativa*.

## Supporting information

**S1 Fig. Comparison of percentage of each amino acid in different PR proteins of *A. thaliana* and *O. sativa*.**
(TIF)

## Author Contributions

**Conceptualization:** Avinash Kaur Nagpal.

**Methodology:** Amritpreet Kaur, Pratap Kumar Pati.

**Supervision:** Pratap Kumar Pati, Aparna Maitra Pati, Avinash Kaur Nagpal.

**Writing – original draft:** Amritpreet Kaur.

**Writing – review & editing:** Pratap Kumar Pati, Aparna Maitra Pati, Avinash Kaur Nagpal.

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
