## [Decision Letter · Decision Letter 0]

24 Jun 2020

PONE-D-20-00611

Physico-chemical Characterization and Topological Analysis of Pathogenesis-related Proteins from Arabidopsis thaliana and Oryza sativa using in-silico approaches

PLOS ONE

Dear Dr. Nagpal,

Thank you for submitting your manuscript to PLOS ONE. After careful consideration, we feel that it has merit but does not fully meet PLOS ONE’s publication criteria as it currently stands. Therefore, we invite you to submit a revised version of the manuscript that addresses the points raised during the review process.

We look forward to receiving your revised manuscript.

Kind regards,

Diego F. Gomez-Casati, PhD

Academic Editor

PLOS ONE

Journal Requirements:

2) Please include captions for your Supporting Information files at the end of your manuscript, and update any in-text citations to match accordingly. Please see our Supporting Information guidelines for more information: http://journals.plos.org/plosone/s/supporting-information.

3)  In your Data Availability statement, you have not specified where the minimal data set underlying the results described in your manuscript can be found. PLOS defines a study's minimal data set as the underlying data used to reach the conclusions drawn in the manuscript and any additional data required to replicate the reported study findings in their entirety. All PLOS journals require that the minimal data set be made fully available. For more information about our data policy, please see http://journals.plos.org/plosone/s/data-availability.

Reviewers' comments:

Reviewer's Responses to Questions

**Comments to the Author**

1. Is the manuscript technically sound, and do the data support the conclusions?

Reviewer #1: Partly

Reviewer #2: Partly

2. Has the statistical analysis been performed appropriately and rigorously? 

Reviewer #1: No

Reviewer #2: No

3. Have the authors made all data underlying the findings in their manuscript fully available?

Reviewer #1: Yes

Reviewer #2: Yes

4. Is the manuscript presented in an intelligible fashion and written in standard English?

Reviewer #1: No

Reviewer #2: Yes

5. Review Comments to the Author

Reviewer #1: Literature review was too shallow and it rarely referenced from pools of already published relevant research. This should be rewritten. The analysis was done without adequate statistical packages to bring home the point. The conclusion contains flying general statements without concluding in specific terms on the results. This conclusion should be rewritten. The entire manuscript should be subjected to editing by native speaker of English language

Reviewer #2: I the view of reviewing manuscript “Physico-chemical Characterization and Topological Analysis of Pathogenesis-related Proteins from Arabidopsis thaliana and Oryza sativa using in-silico approaches ” has been looked thoroughly and observed following points.

1. Author justify that Arabidopsis thaliana and Oryza sativa are different family member of plant so that why compere these of PR proteins.

2. Compare the physico-chemical properties of 12 PR proteins 6 PRs of Arabidopsis

thaliana and 6 PRs of Oryza sativa .while as table no 1, 2 and 3 mentioned only 6 PRs in both Arabidopsis and Oryza PR1, PR2, PR5, PR9, PR10 , PR12 remaining are Figures 1, 2 and 3 why not all 12 PRs mentioned both table and Figures?

3. In introduction section line no.31 Reactive Oxygen Plants (ROS) : Reactive Oxygen Species (ROS).

4. Line no 40-41. many phytohormones like JA, SA and ET: write the full name of JA, SA and ET first time in manuscript.

5. Line No 55- 60 : include appropriate references in introduction section.

6. In table no 1 all entries of At are bold and Os are bold this demarcation pattern create confusion of readers like bold letter have other scientific meaning but this not view of author only differentiate to At and Os. So that table column may shadow.

7. In table no 1. II value of PR 9 and PR10 not stable disused.

8. In table no 2 . Cello, Euloc and Wolfpsort data are not presenting Subcellular localization of PR proteins predicting uniform result .

9. In table no 3. Comparative topological analysis of PR proteins presence or absence of transmembrane domains homogenious resu;t are showing such as TMHMM of PR12 At (1) and Os (0) why.

10. Conclusion should elaborate . Present conclusion no such pin pointing.

11. 5) Kaur A, Pati PK, Pati AM, Nagpal AK . In-silico analysis of cis-acting regulatory elements of

pathogenesis-related proteins of Arabidopsis thaliana and Oryza sativa. PLoS ONE. 2017; 12(9):

e0184523. https://doi.org/10.1371/journal.pone.0184523. Page no Missing.

12. Hofmann K, Stoffel W. TMbase - A database of membrane spanning proteins segments. Biological Chemistry Hoppe-Seyler. 1993; 374:166. doi: citeulike-article-id:9087200. Vol. missing

13. Shen H, Chou JJ. MemBrain: Improving the Accuracy of Predicting Trans membrane Helices. PLoS

ONE. 2008; 3(6): e2399. doi:10.1371/journal.pone.0002399. Page no Missing.

14. No any such current year 2019 and 2020 are present in this manuscript. May some refrences should be add.

15. All figure are not self explainer specially fig. no 3.

6. PLOS authors have the option to publish the peer review history of their article (what does this mean?). If published, this will include your full peer review and any attached files.

Reviewer #1: No

Reviewer #2: No

---

## [Author Response · Author response to Decision Letter 0]

14 Aug 2020

Response to Reviewers

Dear Editor,

Thank you for your mail and valuable suggestions for improvement of our manuscript entitled “Physico-chemical Characterization and Topological Analysis of Pathogenesis-related Proteins from Arabidopsis thaliana and Oryza sativa using in-silico approaches”. We have studied your comments carefully and made major corrections as suggested. The answers to the questions of reviewers are given below.

Journal Requirements:

 Response: Manuscript has been edited according to PLOS ONE’s style.

2) Please include captions for your Supporting Information files at the end of your manuscript, and update any in-text citations to match accordingly.

Response: Caption has been added for Supporting Information file at the end of revised manuscript with track changes (lines 560-562).

3) In your Data Availability statement, you have not specified where the minimal data set underlying the results described in your manuscript can be found. PLOS defines a study's minimal data set as the underlying data used to reach the conclusions drawn in the manuscript and any additional data required to replicate the reported study findings in their entirety. All PLOS journals require that the minimal data set be made fully available. 

Response: The study involves in-silico analysis of 6 PRs of Arabidopsis thaliana and Oryza sativa. In the analysis, the data was tabulated from annotations of Uniprot files of each PR and their accession numbers are given in Table 1.

Reviewer Comments

Reviewer # 1.

1) Query: Literature review was too shallow and it rarely referenced from pools of already published relevant research. This should be rewritten. The analysis was done without adequate statistical packages to bring home the point. The conclusion contains flying general statements without concluding in specific terms on the results. This conclusion should be rewritten. The entire manuscript should be subjected to editing by native speaker of English language.

 Response: The manuscript has been strengthened taking into account the suggestions of the reviewer. Whole introductory part and conclusion have been rewritten completely (pages 2-4, lines 31-99; pages 16-17, lines 331-343, respectively) in the revised manuscript with track changes. Manuscript has been checked for language usage, spelling and grammar by Vasavi Garisetti, Former Language editor, Taylor n Francis, Scientific publishing services, Chennai (India) and changes have been marked in manuscript.

Reviewer # 2

1) Query: Authors justify that Arabidopsis thaliana and Oryza sativa are different family member of plant so that why compere these of PR proteins. 

Response: Arabidopsis thaliana and Oryza sativa are model plants belonging to two different groups i.e. dicots and monocots, respectively. The aim was to determine similarities and differences in PR proteins of these two groups.

2) Query: Compare the physico-chemical properties of 12 PR proteins 6 PRs of Arabidopsis thaliana and 6 PRs of Oryza sativa .while as table no 1, 2 and 3 mentioned only 6 PRs in both Arabidopsis and Oryza PR1, PR2, PR5, PR9, PR10 , PR12 remaining are Figures 1, 2 and 3 why not all 12 PRs mentioned both table and Figures?

 Response: The study relates to 6 PR proteins (PR1, PR2, PR5, PR9, PR10, PR12) each for Arabidopsis thaliana and Oryza sativa. The necessary corrections have been made in the discussion section, in the revised manuscript with track changes (page, 11 and lines 204-205).

3) Query: In introduction section line no.31 Reactive Oxygen Plants (ROS) : Reactive Oxygen Species (ROS).

 Response: Correction have been done in the revised manuscript with track changes (line 40).

4) Query: Line no 40-41. many phytohormones like JA, SA and ET: write the full name of JA, SA and ET first time in manuscript.

Response: Full names are already given in the beginning of the introduction section of the revised manuscript with track changes (lines 41-42).

5) Query: Line No 55- 60 : include appropriate references in introduction section.

Response: These lines highlight the objectives of the present work. (Lines 94-99 in the revised manuscript with track changes).

6) Query: In table no 1 all entries of At are bold and Os are bold this demarcation pattern create confusion of readers like bold letter have other scientific meaning but this not view of author only differentiate to At and Os. So that table column may shadow.

Response: As per the suggestion of the reviewer, alternate columns in table 1 have been shaded, in the revised manuscript with track changes (page 8).

7) Query: In table no 1. II value of PR 9 and PR10 not stable disused.

 Response: Lines 230-231, indicate that PR9 and PR10 of both species; and PR1 and PR12 of A. thaliana only were found to be less than 40 indicating their stable nature.

8) Query: In table no 2 . Cello, Euloc and Wolfpsort data are not presenting Subcellular localization of PR proteins predicting uniform result.

Response: One of the aims of this study was to compare different online tools for subcellular localization of PR proteins and the results are discussed in the discussion section, in the revised manuscript with track changes (page 13, lines 249-261).

9) Query: In table no 3. Comparative topological analysis of PR proteins presence or absence of transmembrane domains homogeneous result are showing such as TMHMM of PR12 At (1) and Os (0) why.

Response: The tool TMHMM indicated the presence of transmembrane domain in AtPR5 as well as AtPR12 but not in OsPR5 and OsPR12. This might be due to some prediction error of the tool itself [1] or may be the transmembrane domain of OsPR5 and OsPR12 did not meet the cutoff of the tool [2]. The same has been added in discussion section in the revised manuscript with track changes (page 15, lines 304-310).

10) Query: Conclusion should elaborate. Present conclusion no such pin pointing.

 Response: Conclusion has been elaborated taking into account the suggestions of the reviewer in the revised manuscript with track changes (page 16-17, lines 331-343).

11) Query: 5) Kaur A, Pati PK, Pati AM, Nagpal AK . In-silico analysis of cis-acting regulatory elements of pathogenesis-related proteins of Arabidopsis thaliana and Oryza sativa. PLoSONE. 2017; 12(9): e0184523. https://doi.org/10.1371/journal.pone.0184523. Page no Missing.

Response: PLOS One uses electronic location identifier instead of page numbers and its already mentioned (e0184523).

12) Query: Hofmann K, Stoffel W. TMbase - A database of membrane spanning proteins segments. Biological Chemistry Hoppe-Seyler. 1993; 374:166. doi: citeulike-article-id:9087200. Vol. missing.

Response: Volume number is already mentioned in the reference, its 374.

13) Query: Shen H, Chou JJ. MemBrain: Improving the Accuracy of Predicting Trans membrane Helices. PLoSONE. 2008; 3(6): e2399. doi:10.1371/journal.pone.0002399. Page no Missing.

Response: PLOS One uses electronic location identifier instead of page numbers and its already mentioned (e2399).

14) Query: No any such current year 2019 and 2020 are present in this manuscript. May some references should be add.

Response: As per suggestion, references for the year 2019 and 2020 have been added in the introduction section, line numbers 37, 62, 66 and 79 and References 1, 2, 13, 17 and 25 have been incorporated in the reference section of the revised manuscript with track changes.

15) Query: All figure are not self explainer specially fig. no 3.

Response: Fig 3 depicts the percentage of secondary structure features (Y axis) of PR proteins (X axis), and different features are shown in different colors as indicated in the figure.

References:

1) Abdullatypov AV. Hup-Type Hydrogenases of Purple Bacteria: Homology Modeling and Computational Assessment of Biotechnological Potential. International journal of molecular sciences. 2020; 21:366. doi: 10.3390/ijms21010366.

2) GENI-ACT; https://doi.org/10.25504/FAIRsharing.mhqkc7; Last accessed: Aug 07, 2020.

---

## [Decision Letter · Decision Letter 1]

15 Sep 2020

Physico-chemical Characterization and Topological Analysis of Pathogenesis-related Proteins from Arabidopsis thaliana and Oryza sativa using in-silico approaches

PONE-D-20-00611R1

Dear Dr. Nagpal,

We’re pleased to inform you that your manuscript has been judged scientifically suitable for publication and will be formally accepted for publication once it meets all outstanding technical requirements.

Kind regards,

Diego F. Gomez-Casati, PhD

Academic Editor

PLOS ONE

Additional Editor Comments (optional):

The manuscript was properly corrected and the authors took into account the suggestions made by the reviewers.

Reviewers' comments:

Reviewer's Responses to Questions

**Comments to the Author**

1. If the authors have adequately addressed your comments raised in a previous round of review and you feel that this manuscript is now acceptable for publication, you may indicate that here to bypass the “Comments to the Author” section, enter your conflict of interest statement in the “Confidential to Editor” section, and submit your "Accept" recommendation.

Reviewer #2: All comments have been addressed

2. Is the manuscript technically sound, and do the data support the conclusions?

Reviewer #2: Yes

3. Has the statistical analysis been performed appropriately and rigorously? 

Reviewer #2: Yes

4. Have the authors made all data underlying the findings in their manuscript fully available?

Reviewer #2: Yes

5. Is the manuscript presented in an intelligible fashion and written in standard English?

Reviewer #2: Yes

6. Review Comments to the Author

Reviewer #2: Author compliances of all my 15 comments and reviewer satisfied with all, specially origin of PR proteins , table formation and rewrite of discussion.

7. PLOS authors have the option to publish the peer review history of their article (what does this mean?). If published, this will include your full peer review and any attached files.

Reviewer #2: No

---

## [Editor Report · Acceptance letter]

17 Sep 2020

PONE-D-20-00611R1

Physico-chemical Characterization and Topological Analysis of Pathogenesis-related Proteins from *Arabidopsis thaliana* and *Oryza sativa* using *in-silico* approaches

Dear Dr. Nagpal:

I'm pleased to inform you that your manuscript has been deemed suitable for publication in PLOS ONE. Congratulations! Your manuscript is now with our production department.

Kind regards,

on behalf of

Dr. Diego F. Gomez-Casati 

Academic Editor

PLOS ONE